# Sustainable Polypropylene Blends: Balancing Recycled Content with Processability and Performance

**DOI:** 10.3390/polym17111556

**Published:** 2025-06-03

**Authors:** Tatiana Zhiltsova, Mónica S. A. Oliveira

**Affiliations:** 1TEMA—Centre for Mechanical Technology and Automation, Department of Mechanical Engineering, University of Aveiro, 3810-193 Aveiro, Portugal; monica.oliveira@ua.pt; 2LASI—Intelligent Systems Associate Laboratory, 4800-058 Guimarães, Portugal

**Keywords:** recycled polypropylene, melt blending, oxidation stability, thermal degradation, mechanical properties, crystallinity, polymer recycling, law of mixtures

## Abstract

The increasing demand for sustainable materials has renewed interest in recycling polyolefins, particularly polypropylene (PP), due to its widespread use and environmental persistence. Post-consumer recycled polypropylene (PP_r_), however, often exhibits compromised properties from prior exposure to thermal, oxidative, and mechanical degradation. This study investigates the potential of using post-consumer PP_r_ in melt-blended extrusion formulations with virgin PP (PP_v_), focusing on how different PP_r_ contents affect processability, thermal stability, oxidative resistance, and mechanical performance. Blends containing 25%, 50%, and 75% PP_r_, as well as 100% PP_r_ and virgin PP, were evaluated using melt flow index (MFI), differential scanning calorimetry (DSC), oxidation induction time (OIT), thermogravimetric analysis (TGA), and tensile testing. Results show that increasing PP_r_ content improves polymer fluidity and thermal stability under inert conditions but significantly reduces oxidation resistance and ductility. However, the 25% PP_r_ blend demonstrated a favourable balance between performance and recyclability, presenting 96% of the elastic modulus and 101% of the yield strength of PP_v_. Homogenization by extrusion improved the oxidative stability of recycled PP by 22% compared to its non-extruded form. These findings support the use of low-to-moderate levels of PP_r_ in virgin PP for applications requiring predictable and tunable performance. contributing to circular economy goals.

## 1. Introduction

The global production of plastics has increased to historically unparalleled levels, reflecting a sustained ascending trend driven by industrial demand and consumer use, with approximately 400 million metric tons (Mt) generated in 2022 alone, of which polypropylene (PP) accounts for nearly 19% of total output [1]. While demand for plastic materials remains high, particularly in packaging, which represented 36.5% of total plastic consumption in 2023, the environmental burden of plastic waste is intensifying [2]. In the same year, an estimated 267.7 Mt of plastic waste was produced globally, yet only 9% was recycled, with the remainder incinerated, landfilled, or mismanaged [1]. A massive amount of this mismanaged waste ends up in the oceans, mainly due to inadequate waste infrastructure in middle-income countries [3]. These developments highlight the importance of implementing circular economy strategies that incorporate mechanical recycling of polyolefins into manufacturing processes, with careful consideration of maintaining their performance in technically demanding applications.

Mechanical recycling constitutes a highly viable strategy for recovering and reintegrating post-consumer plastics into the production cycle. Post-consumer recycled polypropylene, commonly recovered from packaging waste, is often a heterogeneous mixture containing varying grades of PP, stabilizers, and contaminants [4,5,6]. Although attractive from a sustainability perspective, recycled PP is prone to degradation due to previous processing, thermal, mechanical, and UV exposure, leading to reduced molecular weight and overall property deterioration [7,8,9,10,11,12,13]. Even minor chemical modifications can lead to significant alterations in physical properties, particularly when polymers are used in outdoor environments where degradation is triggered by exposure to oxygen and ultraviolet (UV) radiation from sunlight [14,15].

The viability of blending virgin polypropylene with recycled polypropylene has been the focus of several recent investigations. According to Santos et al. [16], incorporating 25% rPP yields tensile strength and modulus values comparable to pure PP, with only slight reductions in elongation at break. Stoian et al. [17] reported that incorporating 50% virgin PP into a recycled PP blend resulted in an 18% increase in tensile strength and a 21% increase in stiffness compared to recycled PP, along with improved thermal stability and crystallinity. In another study, Jamnongkan et al. [18] reported that up to 40 wt.% recycled PP could be added without significantly altering tensile strength or modulus, with slightly higher elongation at break, likely due to increased molecular mobility and better interfacial adhesion during melt processing. Supporting these observations, Hyie et al. [19] identified a 75:25 vPP/rPP ratio as offering a favourable balance between mechanical performance and processability. Recent research has also demonstrated the effectiveness of blending virgin polypropylene with mixed post-consumer polyolefins to tailor performance. Curtzwiler et al. showed that adding up to 100% mixed recycled content (approximately 87.5% PP and 12.5% PE) led to significant improvements in mechanical properties, such as a 74% increase in yield strength and 49% in strain at yield, while also enhancing UV blocking and gas barrier properties. Several of these properties exhibited behaviour consistent with the Law of Mixtures, allowing for controlled adjustment of material performance through deliberate changes in blend composition. These findings underline the complex behaviour of vPP/rPP blends, where mechanical and thermal properties do not always change in a linear fashion. Variability in recycled content, including the presence of inorganic fillers or residual additives, can affect crystallinity, density, and phase morphology, which in turn influence stress distribution and failure mechanisms. Hinczica et al. [20] also pointed out that while 10% rPP had minimal impact on yield strength and resistance to crack growth, higher proportions led to a notable decline in slow crack growth (SCG) performance and impact resistance. Their work highlights the role of molecular weight distribution and contaminant levels in determining long-term durability. Despite these limitations, recycled PP remains attractive due to its environmental advantages and lower cost compared to virgin material. It is generally more affordable than virgin polymers and helps reduce reliance on fossil-derived resources. Hyie et al. [19] further emphasized that maintaining consistent processing conditions, particularly in drying, homogenizing, and pelletizing, is key to achieving reliable mechanical behaviour when using post-consumer recycled materials.

Although the blending of virgin and recycled polypropylene has been widely studied, most research relies on commercially sourced recycled PP with unknown or inconsistent composition, often containing contaminants such as other polymers [8,11,13,17,19,20], which introduces uncertainty and limits the reproducibility of results. These contaminants can alter the properties of the blends, making it challenging to establish structure–property relationships.

In contrast, the novelty of this study lies in the use of post-consumer PP sourced from a decentralized recycling stream with a fully documented processing history, ensuring the absence of contamination with other polymer types. Additionally, this work investigates whether the mechanical, thermal, and oxidative properties of virgin/recycled PP blends across various ratios (25%, 50%, 75%) follow the Law of Mixtures (LOM [21,22]), a concept not systematically validated for recycled PP systems. Furthermore, the study compares extruded and non-extruded recycled PP, obtained from the authors’ previous work [23], to assess the effect of melt homogenization on material performance. These aspects provide new insights into the predictability and optimization of recycled PP blends for more performance-demanding applications.

## 2. Materials and Methods

### 2.1. Materials

The virgin PP_v_ (CAPILENE R 50 homopolymer supplied by Carmel Olefins (Haifa Bay, lsrael)) was used as a reference material, and for compounding with post-consumer PP, designated thereafter as PP_r_, was gathered from various decentralized recycling points through the yellow bin collection system. PP_r_ was shredded and washed in cold water. More information about the preparation of PP_r_ may be found elsewhere [23].

### 2.2. Methods

#### 2.2.1. Preparation of the PP_v_/PP_r_ Blends

Prior to extrusion, PP_v_ and PP_r_ were mechanically mixed based on their weight percentage. The mixing rates were selected to systematically evaluate the effect of increasing recycled content in a controlled manner and may be consulted in Table 1. This approach is consistent with previous studies on post-consumer PP recycling blends carried out by Santos et al. [16] and Badini et al. [24]. Although polypropylene is a non-hygroscopic polymer, both virgin and post-consumer recycled PP were dried at 80 °C for 2 h prior to extrusion in the Meemert Universal Oven UF30 (Memmert GmbH + CO.KG, Schwabach, Germany) to eliminate any surface moisture and minimize processing inconsistencies. The premix was fed at a constant screw speed of 12 rpm using an 11 mm single-screw feeder (Thermo Electron GmbH, Karlsruhe, Germany) connected to a Thermo Scientific™ Process 11 co-rotating twin-screw extruder (11 mm diameter, L/D ratio 40:1; Thermo Fisher Scientific Inc., Waltham, MA, USA). The screw configuration followed a standard three-stage design. It included conveying elements for material transport, as well as a combination of distributive and dispersive mixing elements, such as kneading blocks and reverse elements, to promote homogeneous blending of virgin and recycled PP fractions.

Compounding was carried out at a screw speed of 100 rpm, with the temperature gradually increasing across the extruder barrel’s seven zones from the hopper to the 2 mm die, set at 180 °C, 185 °C, 190 °C, 195 °C, 200 °C, 210 °C, and 220 °C, respectively. Five batches were prepared, as detailed in Table 1. Virgin PP was used in pellet form, as received, while PP_r_, originally in the form of shredded flakes, was extruded separately to ensure consistent processing conditions across all PP_v_ and PP_r_ blends. The extruded PP_r_ was then granulated using an SM 100 Cutting Mill (RETSCH GmbH, Haan, Germany) equipped with a 4 mm mesh sieve, yielding granules for further testing and processing.

#### 2.2.2. Preparation of Tensile Test Specimen

Tensile test specimens (ISO 527-2 type 5A [25]) were produced by injection moulding using a HAAKE MiniJet II mini-injection moulding machine (Thermo Fisher Scientific, Waltham, MA, USA). The processing conditions used for virgin PP, recycled PP, and their blends are detailed in Table 2.

#### 2.2.3. Melt Flow Index Assessment

The melt flow index was measured using a Göttfert MI-3 machine (GÖTTFERT Werkstoff-Prüfmaschinen GmbH, Buchen, Germany) in accordance with ISO 1133-1997 (2.16 kg, 230 °C) [26]. For each material, five samples were tested to ensure accuracy and reliability.

#### 2.2.4. Thermal Analysis

To determine the melting temperature (T_m_), crystallization temperature (T_c_), melting enthalpy (H_m_), and cold crystallization enthalpy (H_c_), differential scanning calorimetry (DSC) was performed on samples weighing approximately 5.5 ± 0.5 mg. The tests were conducted using a DSC Discovery 250 instrument (TA Instruments, New Castle, DE, USA) by ISO 11357-6:2018 [27]. Data were analysed using TRIOS software (V5.7), developed by TA Instruments. To eliminate any thermal history, each material lot underwent two heating and cooling cycles, with data from the second cycle used for analysis. Three samples were tested per material lot. The DSC procedure involved stabilizing each sample at 20 °C, heating it to 200 °C, and then cooling it back to 20 °C at a rate of 10 °C/min. The degree of crystallinity (χ) was calculated using Equation (1) [28](1)$$\chi \left(\%\right)=\frac{H_{m}}{H_{m}^{0}}\times 100$$
where H_m_ (J/g) represents the melting enthalpy of the polymer under analysis. The melting enthalpy of 100% crystalline PP ($H_{m}^{0}$) is known to be 207 J/g [29].

The oxidative resistance of the materials was evaluated according to ASTM D3895-14 [30]. Film samples weighing approximately 7 ± 1.0 mg were placed in open aluminum crucibles, forming a uniform layer at the bottom. The samples were first heated under a nitrogen atmosphere from room temperature to 200 °C at a rate of 20 °C/min, then held isothermally for 5 min. After that, the nitrogen atmosphere was replaced with oxygen and maintained until the onset of the exothermic peak.

Thermal stability was assessed by thermo-gravimetric analysis (TGA), performed in a Netzsch–Jupiter STA 449 F3 apparatus (NEDGEX GmbH, Selb, Germany) according to E 2550-11 [31]. Samples weighing 10 ± 1 mg were taken from the extruded PP_r_ and its blends with PP_v_, while the PP_v_ sample was obtained from pellets. The samples were heated from 30 °C to 700 °C at a constant rate of 20 K/min under a nitrogen atmosphere (50 mL/min) in alumina (Al_2_O_3_) crucibles, and weight loss was recorded as a function of temperature. The derivative thermogravimetric (DTG) curve, representing the rate of weight change, was also used to help interpret thermal degradation events.

#### 2.2.5. Mechanical Properties

Tensile tests were conducted using a Shimadzu AGS-X 10 kN machine (Shimadzu Scientific Instruments, Columbia, MD, USA) by ISO 527-1 [32]. All tests were performed at room temperature in two stages. In the first stage, specimens were stretched at a 1 mm/min rate to determine Young’s modulus. The tensile rate was increased to 50 mm/min in the second stage and maintained until specimen failure. Data from this stage were used to calculate the yield stress (σ_y_) and strain (ε_y_), as well as the tensile strength (σ_b_) and strain at break (ε_b_). Five specimens were tested for each material batch.

## 3. Results and Discussion

### 3.1. Melt Flow Index

The melt flow index measurements revealed a clear trend of increasing fluidity with the addition of post-consumer recycled polypropylene to virgin polypropylene. PP_v_ exhibited the lowest MFI value (11.94 ± 0.53 g/10 min), while 100% recycled PP showed the highest MFI (40.72 ± 2.16 (g/10 min)). This trend reflects a corresponding decrease in molecular weight, consistent with expectations for virgin blends of high MW (molecular weight) and recycled PP with low MW. No significant difference in the MFI (40.56 ± 1.49 (g/10 min)) was detected for the material (rPPcw) investigated in our earlier work [23] when compared to PP_r_. The rPPcw is identical in its composition to PP_r_. The main difference is that the latter underwent an additional extrusion step to ensure the homogenization of the recycled plastics mixture, while the rPPcw was moulded directly from reground recycled flakes.

Intermediate compositions, containing 25%, 50%, and 75% PP_r_, demonstrated a progressive increase in MFI values (17.74 ± 0.61, 23.32 ± 0.98, and 29.96 ± 0.66 (g/10 min), respectively) and hence a decrease in molecular weight [33]. Verification of the results via regression analysis (Figure 1) led to the conclusion that the MFI of the blends follows the Law of Mixtures (LOM) with good prediction reflected in the coefficient of determination of the fit *R*^2^ = 0.9788. The Law of Mixtures states that the properties of binary mixtures fall between those of the individual pure components and vary proportionally with their respective volume fractions (ν) [21,22]. In the case of melt flow index, the MFI of a blend (*MFI_b_*) should vary according to Equation (2), where subscripts *v* and *r* denote the virgin and recycled states, respectively.(2)$$MFI_{b}=MFI_{v}\times \vartheta_{v}+MFI_{r}\times \vartheta_{r}$$

While the increase in MFI is partially due to the presence of lower molecular weight fractions in PP_r_ (which originates from mixed-grade PP with a broad MFI range), prior thermal and mechanical degradation likely contributes to further molecular weight reduction. It must be taken into account, as lower molecular weight typically results in diminished mechanical performance, including reduced tensile strength, impact resistance, and elongation at break [4,7,8,34,35,36].

### 3.2. Thermal Characterization

#### 3.2.1. Thermal Transitions and Crystallinity Analysis

The thermal behaviour of PPᵥ and its blends with PP_r_ was evaluated by DSC, and the respective values are exemplified in Figure 2 and may be consulted in Table 3, where H_c_ (J/g) and T_c_ (°C) are, respectively, crystallization enthalpy and temperature; H_f_ (J/g) and T_f_ (°C) are, respectively, fusion enthalpy and temperature; and χ (%) is the crystallinity degree. Table 4 shows how the thermal properties of the blends and PP_r_ vary in comparison to PP_v_.

The fusion enthalpy decreased slightly with increasing PP_r_ content, from 101.59 ± 2.06 (J/g) for PPᵥ to 92.04 ± 1.37 (J/g) for 100% PP_r_, accompanied by a minor reduction in the degree of crystallinity (from 49.08% to 44.46%). However, the fusion temperature remained relatively stable across all compositions, ranging from 163.61 ± 0.12 °C (PPᵥ) to 162.63 ± 0.09 °C (PP_r_), indicating that the crystalline structure of polypropylene does not undergo significant alteration. Baldini et al. [24] reported similar melting behaviour; however, their study observed a double crystallization peak during cooling, attributed to polyethylene contamination, common in post-consumer polymer waste.

It should be noted, however, that in the present study, the blend containing 25% recycled polypropylene (25%PP_r_) exhibited slightly higher melting temperature, melting enthalpy, and hence, degree of crystallinity compared to the PP_v_ (Table 3 and Table 4). This behaviour can be attributed to residual crystalline fragments and particulate contamination within the recycled material, which may act as heterogeneous nucleating agents when blended at a relatively low proportion (25%) with virgin PP [37]. These nucleating sites facilitate the crystallization process by promoting faster and more efficient organization of polymer chains during cooling. Furthermore, slight chain scission occurring during the recycling process may enhance molecular mobility, thereby favouring crystallization at low content of the recycled PP [38]. These combined effects result in an initial increase in crystallinity at low PP_r_ content before the negative impact of accumulated degradation becomes more pronounced at higher PP_r_ percentages. In contrast, Stoian et al. [17] reported a slight increase in crystallinity with increasing amounts of recycled PP derived from raffia. This discrepancy may be attributed to the inherently higher crystallinity of the raffia-based rPP compared to the virgin PP used in their study.

During the cooling cycle, the recrystallization enthalpy (H_c_) decreased with higher PP_r_ content. Notably, the recrystallization temperature (T_c_) of the blends increased approximately 10%, from 112.3 °C for PPᵥ to 124.2 °C for PPᵣ. This shift suggests that the recycled material promotes earlier nucleation during cooling, likely due to the presence of heterogeneous nucleation sites generated during prior processing cycles [39]. This trend is consistent with observations by Infurna et al. [40], where the crystallization temperature increase with the gradual addition of PPr was attributed to a higher proportion of shorter polymeric chains.

Overall, the incorporation of recycled post-consumer polypropylene into virgin material does not significantly affect the thermal transitions and crystallinity. It is also worth noting that, as shown in Figure 2, virgin PP, PP_r_, and their blends each exhibit a single melting and crystallization peak with no additional thermal transitions, indicating high purity across all samples. However, a slightly asymmetric recrystallization peak, observed for the 25% PP_r_ blend (Figure 2), suggests a more complex nucleation process. This asymmetry may result from the moderate recycled content acting as a nucleating agent, promoting multiple crystallization events. Such behaviour could indicate the formation of a finer and more heterogeneous crystalline morphology, driven by the presence of the recycled fraction. Additionally, the slight increase in crystallinity suggests a potential improvement in mechanical properties, which will be confirmed upon further assessment.

The comparison between the thermal properties of extruded recycled polypropylene (PP_r_) and its non-extruded counterpart (rPPcw) [23] highlights the influence of prior homogenization on crystallization behaviour. While both materials show similar melting temperatures, rPPcw exhibits a higher melting enthalpy (99.05 ± 0.54 (J/g)) and crystallinity (48.00%) compared to PP_r_ (92.04 ± 1.37 (J/g); 44.46%). This increase in crystallinity may be attributed to residual ordered structures or crystal fragments preserved in the rPPcw due to the absence of additional thermal and shear processing steps during extrusion [37,39]. The crystallization temperature (T_c_) of rPPcw is slightly lower (122.23 °C) than that of PP_r_ (124.19 °C), suggesting that although the crystallinity is higher, nucleation may have occurred later or more gradually. This behaviour may result from less uniform chain distribution or incomplete homogenization of rPPcw, as the material was directly processed from reground flakes. These findings suggest that the homogenization by extrusion process can disrupt some of the existing crystalline domains, reducing overall crystallinity and sequential stiffness of the recycled PP but improving its ductility, as it was verified by mechanical testing further discussed ahead.

#### 3.2.2. Oxidation Induction Time

Polypropylene is prone to oxidative degradation due to tertiary carbon atoms in its backbone, which are easily attacked by free radicals, especially under heat or UV exposure. It leads to chain scission, resulting in embrittlement and loss of mechanical properties [41,42]. Therefore, OIT analysis is essential for understanding how blending with virgin PP can attenuate oxidative degradation.

As shown in Table 5 and exemplified in Figure 3a, virgin polypropylene’s oxidation induction time was 277.6 ± 55.3 s, while adding just 25% PP_r_ led to a sharp decrease to 81.3 ± 6.9 s, representing a reduction of approximately 71%. Further increases in PP_r_ content continued to reduce the OIT, with values of 75.4 ± 7.4 s, 64.0 ± 6.9 s, and 50.8 ± 8.3 s for 50%, 75%, and 100% PP_r_, respectively. These results indicate that recycled polypropylene contains a significantly lower amount of antioxidant stabilizers, which are typically consumed during the material’s prior thermal and mechanical processing cycles. The loss of antioxidants and the existence of oxidative breakdown products like peroxides and carbonyl compounds lower the material’s resistance to further oxidation attack [23,37].

Furthermore, recycled PP may contain residual catalysts and metal contaminants, which can act as pro-oxidants, enhancing the degradation rate. Such impurities exacerbate thermo-oxidative degradation of PP [43]. The presence of small amounts of previously degraded material may also trigger localized catalytic effects, accelerating the onset of oxidation and contributing to a reduced OIT. Therefore, OIT should be interpreted as a relative indicator of oxidative stability under controlled conditions rather than a direct predictor of long-term service life.

The slight improvement in oxidation induction time (OIT) from 41.6 ± 5.5 s [23] in non-extruded recycled polypropylene (rPPcw) to 50.8 ± 8.3 s in extruded PP_r_ can be attributed to the homogenization effect of the extrusion process. Extrusion promotes a more uniform distribution of oxidative degradation products and residual antioxidants, reducing localized concentrations of reactive species that would otherwise accelerate degradation. Additionally, thermal and shear conditions during extrusion may partially remove volatile oxidative byproducts and facilitate structural reorganization, slightly enhancing oxidative resistance even without added stabilizers [44].

Besides the sharp reduction in oxidation induction time (OIT) observed with the incorporation of 25% post-consumer recycled polypropylene (PP_r_), the relationship between OIT and recycled content remains linear across the 25% to 100% range (Figure 3b) in line with the Law of Mixtures, highlighting the strong negative correlation between increasing PP_r_ content and decreasing OIT at this range.

#### 3.2.3. Thermogravimetric Analysis

As demonstrated in Figure 4 and Table 6, the initial degradation temperature (T_on_) of the virgin polypropylene is 325.5 °C, whereas all blends containing recycled PP exhibit a substantially higher T_on_ around 344 °C (for 25%, 50%, 75% PP_r_ blends and 100% PP_r_). PP_r_ exhibits significantly higher T_on_ values, around 344 °C. This represents an increase in approximately 19 °C with the addition of any recycled content, indicating enhanced thermal stability of the blends and PP_r_ under an inert atmosphere. A similar trend has been reported by Stoian et al., who also reported higher degradation onset in recycled PP and its blends with virgin PP compared to the latter [17].

Capilene, as a commercial-grade polypropylene homopolymer intended for general-purpose applications, is expected to contain a typical antioxidant (AO) package, primarily to protect against oxidative degradation during melt processing and service life. However, recycled PP contains a mixture of polypropylene grades from unknown sources and various end-use applications and may retain residual antioxidants and stabilizers from inherently more stable grades in the recycled mix that delay the onset of thermal degradation [41]. The recycled fraction imparts enhanced stability to the blends: even at 25% PP_r_ content, elevating T_on_ to the same level as 100% recycled PP, suggesting the presence of stabilizers or altered polymer segments in the recycled material dominates the initial decomposition behaviour of the blend, outweighing the crystallinity loss (Table 3 and Table 4).

The peak degradation temperature (T_p_) corresponds to the temperature of the fastest mass loss (the DTG peak), as shown in detail in the inset of Figure 4b. All samples show a single prominent degradation peak, with T_p_ values clustered in the mid-460 °C range. Virgin PP presents a T_p_ of 463.3 °C, and the T_p_ of the 25–75% PP_r_ blends are very similar (within ±1 °C, around 462–465 °C). It indicates that the primary decomposition event occurs at essentially the same temperature for virgin and recycled blends, implying that the fundamental degradation mechanism is unchanged and typical of polypropylene, known to undergo a one-step, random-chain scission thermal decomposition [38]. Moreover, it also indicates a single-step degradation for all formulations (virgin, recycled, and blends). The presence of recycled content, up to 75%, did not significantly alter the kinetics of this degradation step. The fully recycled PP shows a slightly higher T_p_ (469.3 °C), about 6 °C above the virgin/blend values, indicating higher thermal stability. This slight shift to a higher peak temperature for the recycled sample implies that once degradation starts, minor cross-linking/branching from its past lifecycle or trace contaminants in the recycled stream, which decomposes at higher temperatures, may influence the peak position [45]. Moreover, recycled PP may retain residual stabilizers such as phenolic antioxidants from previous processing cycles, which can delay thermal decomposition and increase T_p_ under inert conditions [44].

However, the thermal stability observed under inert conditions in TGA analysis does not reflect the oxidative stability of recycled PP and its blends with virgin PP. The oxidation induction time (OIT) results presented in Table 5 show a clear and significant decline in oxidative resistance with increasing recycled content. While PP_r_ exhibits a higher T_on_ and T_p_ in TGA, it also shows an 82% reduction in OIT compared to PP_v_, indicating that it has undergone oxidative degradation during its prior use and recycling. This degradation is likely caused by chain scission and the formation of oxidation-prone functional groups, such as carbonyls [43]; however, the latter degradation mechanism is mostly absent under nitrogen in TGA analysis. Therefore, while recycled PP thermal decomposition is delayed at higher temperatures under inert conditions, its chemical structure makes it more vulnerable to oxidative attack [38,44,46].

Furthermore, the degradation range (ΔT = T_e_ − T_p_), which represents the temperature interval over which most decomposition occurs, supports this interpretation. For virgin PP, T_e_ (end degradation temperature) is about 497.6 °C, resulting in a broad degradation interval of 172.1 °C. In contrast, the blends and recycled PP exhibit slightly higher T_e_ values (501–502 °C) but narrower ΔT ranges −ΔT of approximately 156–158 °C, degrading over a narrower temperature range, likely due to the presence of thermally unstable regions, such as oxidized or low molecular weight segments, resulting from prior thermal and mechanical degradation, and leading to a faster thermal decomposition process.

### 3.3. Mechanical Characterization

The tensile test results for virgin polypropylene (PP_v_), its blends with post-consumer recycled polypropylene (PP_r_), and unprocessed recycled PP (rPPcw) provide insights into how increasing recycled content affects mechanical properties (Table 7 and Figure 5). To enable a comparison between the different mechanical properties, all values were normalized relative to PP_v_ (Figure 5b). There is a clear decreasing trend in elastic modulus (E) with increasing PP_r_ content. PP_v_ exhibits the highest modulus at approximately 1378 MPa, while 100% PP_r_ shows a reduced modulus of about 984 MPa. This 29% decline is due to the degradation of polymer chains during recycling, leading to lower molecular weight and reduced stiffness [39]. As shown in Figure 6, the elastic modulus follows the Law of Mixtures, allowing for accurate prediction of how PP_v_/PP_r_ will perform in the elastic deformation region.

The yield strength (σ_y_) remains relatively stable across the blends, with a maximum of 15% reduction from 36.67 MPa in PP_v_ to 31.08 MPa in PP_r_. It indicates that the initial resistance to plastic deformation is not significantly compromised by the addition of recycled content, possibly due to the retention of crystalline regions that contribute to yield behaviour [44]. These findings corroborate the gradual decrease in elastic modulus and yield strength reported by Hincza et al. [20] in virgin and recycled PP blends, not specifying, however, the origin of recycled PP. Curtzwiler et al. [11] reported an opposite trend, showing a linear increase in yield strength and strain with increasing content of post-consumer polyolefin recyclates. These contradictory findings, however, may be explained by the composition of the recycled post-consumer material used by these authors, which consisted mainly of polyethylene with a low proportion of polypropylene.

Ultimate tensile strength (σ_b_) and elongation at break (ε_b_) show a marked decrease with higher PP_r_ content. PP_v_ has an ultimate tensile strength of 22.80 MPa and elongation at break of 293.86%, whereas PPr drops to 3.30 MPa and 109.79%, respectively. As shown in Figure 5b, both elongation at break and break strength display non-linear trends with increasing PP_r_ content. While a consistent decrease might be expected due to degradation of the recycled phase, the partial recovery of both ε_b_ and σ_b_ at 75% PPr suggests that morphological rearrangement or improved phase continuity may occur at higher recycled content. This could enhance stress distribution or local chain mobility within the matrix. However, at 100% PP_r_, the effects of oxidative and thermal degradation dominate, leading to severe embrittlement and loss of mechanical integrity, as reflected by the 63% reduction in break strength and 85% reduction in elongation at break. The reduction in these properties suggests that chain scission and previously mentioned oxidative degradation during recycling reduce the material’s ability to undergo plastic deformation before failure.

Unprocessed by extrusion, rPPcw (Table 7) from these authors’ previous work [23] exhibits an elastic modulus comparable to PP_v_ (1379.72 MPa) and a higher ultimate tensile strength (23.32 MPa) but significantly lower elongation at break (18.75%). This behaviour suggests that while stiffness and strength may be retained, the lack of homogenization through extrusion leads to poor ductility, possibly due to contaminants and inhomogeneities acting as stress concentrators [37]. Overall, the incorporation of recycled polypropylene affects the mechanical properties of the blends, with higher recycled content leading to reduced stiffness, strength, and ductility. However, at lower concentrations (25% PP_r_), the impact on mechanical properties is less pronounced and even slightly improved, as is the case with yield strength, indicating the potential for using recycled content without significantly compromising material performance.

## 4. Conclusions

This study confirms that blending virgin polypropylene with post-consumer recycled polypropylene, sourced from decentralized recycling systems, allows for controlled tuning of thermal, oxidative, and mechanical properties. The melt flow index (MFI), elastic modulus, and partially OIT followed the Law of Mixtures, enabling predictable property evolution across blend ratios. The 25% PP_r_ blend exhibited the best balance of properties, such as improved crystallinity, yield strength, and thermal stability. In contrast, higher recycled content led to a significant decline in oxidation resistance, with OIT decreasing by 82% at 100% PPr, indicating the need to reintroduce stabilizers. Extrusion-induced homogenization improved oxidative stability (22%) and ductility compared to non-extruded recycled material. These findings support the practical use of low-to-moderate recycled PP levels in non-critical applications and contribute new insight into the predictability and processability of recycled PP blends. Future work should evaluate long-term durability and the effects of additive re-stabilization.

## Figures and Tables

**Figure 1 polymers-17-01556-f001:**
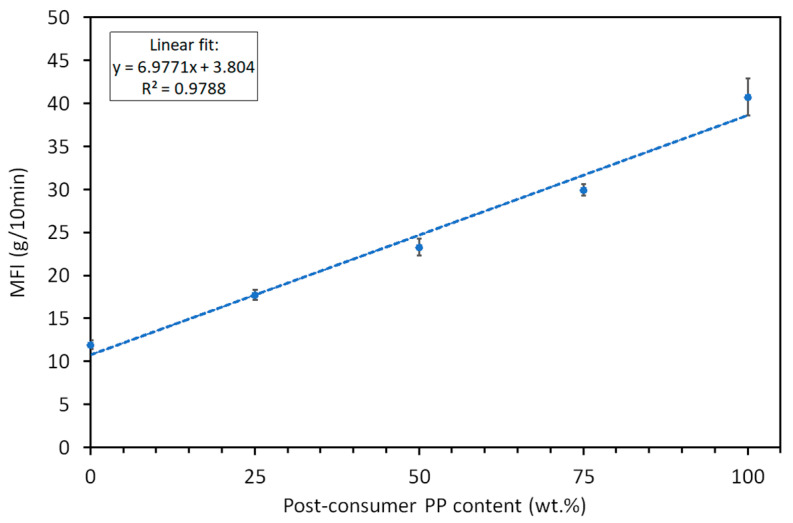
MFI of PP_v_, PP_r_, and their blends at different recycled content levels. The dotted line represents the linear regression fit.

**Figure 2 polymers-17-01556-f002:**
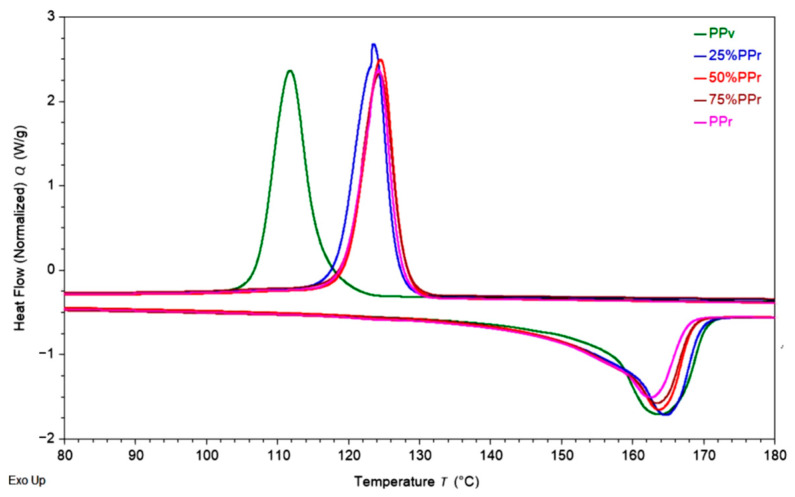
DSC thermograms of PP_v_, PP_r_, and their blends at different recycled content levels.

**Figure 3 polymers-17-01556-f003:**
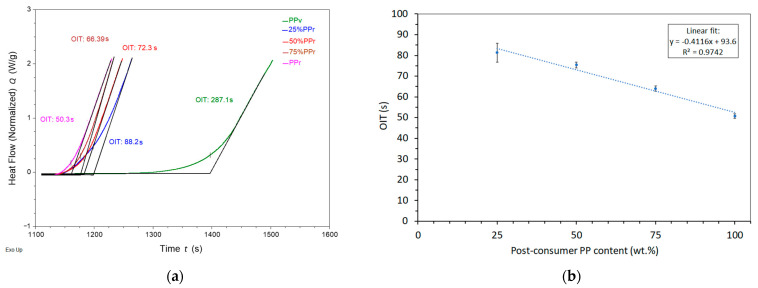
Oxidation induction time (OIT) of PP_v_ and its blends with PP_r_: (**a**) Full dataset including virgin PP; (**b**) PP_v_/PP_r_ blends at different PP_r_ recycled content. The dotted line represents the linear regression fit.

**Figure 4 polymers-17-01556-f004:**
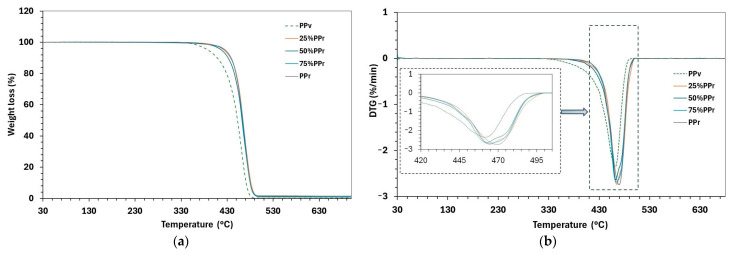
Thermal properties of PP_v_, PP_r_, and their blends (**a**) TGA thermograms (**b**) Derivative thermogravimetry thermograms. The inset in (**b**) highlights the differences in peak degradation temperatures among the samples.

**Figure 5 polymers-17-01556-f005:**
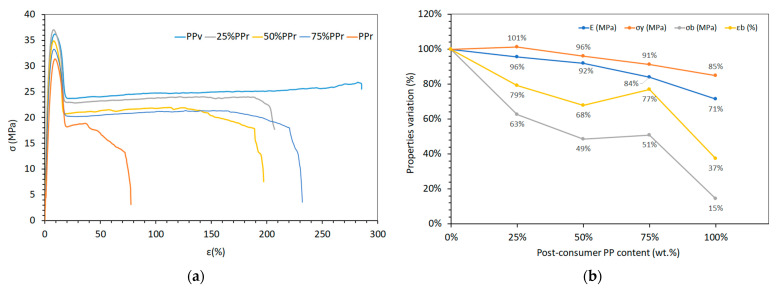
Tensile properties of PP_v_, PP_r_, and their blends: (**a**) Stress–strain curves; (**b**) normalized mechanical properties (%).

**Figure 6 polymers-17-01556-f006:**
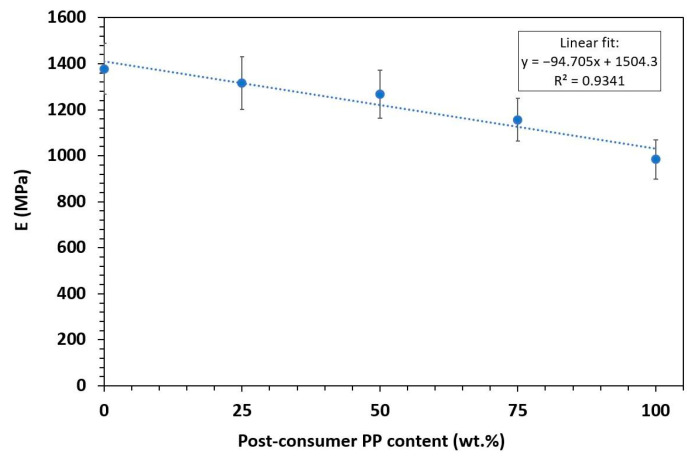
Elastic modulus of PP_v_, PP_r_, and their blends at different PP_r_ recycled content. The dotted line represents the linear regression fit.

**Table 1 polymers-17-01556-t001:** Composition of the PP batches.

Batch	PP_v_ (wt.%)	PP_r_ (wt.%)
PP_v_	100	0
25%PP_r_	75	25
50%PP_r_	50	50
75%PP_r_	25	75
PP_r_	0	100

**Table 2 polymers-17-01556-t002:** Injection molding processing conditions.

Processing Parameter	Value
Plasticization chamber temperature (°C)	230
Mold temperature (°C)	40
Injection time (s)	3
Injection pressure (bar)	300
Packing time (s)	17
Packing pressure (bar)	240

**Table 3 polymers-17-01556-t003:** Thermal properties of PPv, PPr, and their blends, determined by DSC.

Material Composition	H_c_ (J/g)	T_c_ (°C)	H_f_ (J/g)	T_f_ (°C)	χ (%)
PP_v_	102.16 ^1.^ (±) 1.89	112.25 (±) 0.37	101.59 (±) 2.06	163.61 (±) 0.12	wi
25%PPr	99.60 (±) 0.36	123.68 (±) 0.39	103.60 (±) 0.75	164.44 (±) 0.19	50.05
50%PPr	94.23 (±) 1.61	124.41 (±) 0.10	97.37 (±) 3.50	163.75 (±) 0.07	47.04
75%PPr	92.04 (±) 0.73	124.30 (±) 0.12	95.40 (±) 0.71	163.30 (±) 0.07	46.09
PP_r_	86.82 (±) 0.58	124.19 (±) 0.02	92.04 (±) 1.37	162.63 (±) 0.09	44.46
^2.^ rPPcw	^3.^ -	122.23 (±) 0.16	99.05 (±) 0.54	162.32 (±) 0.21	48.00

^1.^ (±) represents the standard deviation. ^2.^ data from the study by Prior et al. [23] are included for comparison purposes. ^3.^ - data were not available.

**Table 4 polymers-17-01556-t004:** Variation of thermal properties (%) of PP blends and PP_r_ relative to virgin PP.

Material Composition	^1.^ ∆H_c_ (%)	∆T_c_ (%)	∆H_f_ (%)	∆T_f_ (%)
25%PP_r_	^2.^ ↓ 2.51	^3.^ ↑ 10.18	↑ 1.98	↑ 0.51
50%PP_r_	↓ 7.76	↑ 10.83	↓ 4.15	↑ 0.09
75%PP_r_	↓ 9.91	↑ 10.73	↓ 6.09	↓ 0.19
PP_r_	↓ 15.02	↑ 10.64	↓ 9.40	↓ 0.60

^1.^ ∆ variation. ^2.^ ↓ reduction. ^3.^ ↑ increase.

**Table 5 polymers-17-01556-t005:** Oxidation Induction Time of PP_v_, PP_r_, and their blends.

Material Composition	OIT (s) ^1.^ (±)
PP_v_	277.6 (±) 55.3
25%PP_r_	81.3 (±) 6.9
50%PP_r_	75.4 (±) 7.7
75%PP_r_	64.0 (±) 6.9
PP_r_	50.8 (±) 8.3
^2.^ rPPcw	41.6 (±) 5.54

^1.^ (±) represents the standard deviation. ^2.^ data from the study by Prior et al. [23] are included for comparison purposes.

**Table 6 polymers-17-01556-t006:** Thermogravimetric Analysis of PP_v_, PP_r,_ and their Blends.

Material Composition	T_on_ (°C)	T_e_ (°C)	T_p_ (°C)	ΔT = T_e_ − T_p_ (°C)
PP_v_	325.5	497.6	463.3	172.1
25%PP_r_	345.0	501.0	464.5	156.0
50%PP_r_	344.2	501.8	462.6	157.6
75%PP_r_	344.1	501.9	464.0	157.8
PP_r_	344.2	500.8	469.3	156.6

**Table 7 polymers-17-01556-t007:** Mechanical properties of PP_v_, PP_r,_ and their blends.

Material Composition	E (MPa)	σy (MPa)	σb (MPa)	εb (%)
PP_v_	1377.95 ^1.^ (±) 109.99	36.67 (±) 0.93	22.80 (±) 3.39	293.86 (±) 21.00
25%PP_r_	1315.45 (±) 114.16	37.16 (±) 0.32	14.30 (±) 4.16	233.07 (±) 24.99
50%PP_r_	1266.98 (±) 103.95	35.22 (±) 0.41	11.07 (±) 3.83	199.37 (±) 30.40
75%PP_r_	1156.32 (±) 92.92	33.48 (±) 0.26	11.59 (±) 4.67	226.32 (±) 32.37
PP_r_	983.99 (±) 86.11	31.08 (±) 0.34	3.30 (±) 1.21	109.79 (±) 21.90
^2.^ rPPcw	1379.72 (±) 59.13	33.90 (±) 0.70	23.32 (±) 4.45	18.75 (±) 4.17

^1.^ (±) represents the standard deviation. ^2.^ data from the study by Prior et al. [23] are included for comparison purposes.

## Data Availability

The original contributions presented in this study are included in the article. Further inquiries can be directed to the corresponding author.

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
