# Peer review of "Sustainable Polypropylene Blends: Balancing Recycled Content with Processability and Performance"

_polymers, 2025, doi:10.3390/polym17111556_

Round 1
Reviewer 1 Report
Comments and Suggestions for Authors
- The novelty of the current work must be highlighted in the current work. Since there are other works related mixing virgin and recycled PP.
- More critical findings from the current work should be highlighted in the abstract. It needs some quantitative results.
- Why is Eb at 75% higher while it is lower for 50% PP content.
- How many heating zones were present in the twin screw extruder. The manufacturing procedure must be elaborated.
- Suggested to include the DSC curved. The crystallization behavior and melting point must be presented
- The outcomes is not clear from the results, it is suggested to concise the conclusion.
Author Response
Dear Reviewer,
The authors sincerely thank you for your critical comments, helpful suggestions, and, most importantly, for dedicating your valuable time to review our manuscript. We have carefully considered all your comments and suggestions and believe that the revised version has been improved as a result. Please find attached a point-by-point response to your comments.
Comment 1. The novelty of the current work must be highlighted in the current work. Since there are other works related mixing virgin and recycled PP.
Response 1: Thank you for the pertinent comment. In the revised manuscript, we have highlighted the novelty of our work in the Introduction section (lines 91 – 106)
Comment 2: More critical findings from the current work should be highlighted in the abstract. It needs some quantitative results.
Response 2: The abstract has been revised to include the key quantitative findings (lines 24-26). Specifically, the authors highlighted the change of 96% of the elastic modulus and 101% of the yield strength in the 25% PPr blend compared to PPv. These additions aim to reflect better the critical outcomes and practical implications of our study.
Comment 3: Why is Eb at 75% higher while it is lower for 50% PP content.
Response 3: Thank you for pointing this out. This point has been addressed with additional clarification in the Results and Discussion section (lines 431-443). As shown in Figure 5(b), both elongation at break and break strength display non-linear trends with increasing PPr content. While a steady decrease in εb might be expected due to the degraded nature of the recycled phase, the partial recovery observed at 75% PPr likely results from morphological rearrangement or improved phase continuity at higher recycled content. This could facilitate better stress distribution and enhance local chain mobility, leading to increased ductility. At 100% PPr, however, degradation effects dominate, causing severe embrittlement.
Comment 4: How many heating zones were present in the twin screw extruder. The manufacturing procedure must be elaborated.
Response 4: The authors have updated the Materials and Methods section (lines 117 and 140) to include details of the extrusion setup. The Thermo Scientific Process 11 twin-screw extruder was equipped with seven heating zones, with the temperatures progressively set from the hopper to the die at 180 °C, 185 °C, 190 °C, 195 °C, 200 °C, 210 °C, and 220 °C to ensure effective melt blending and homogenization.
Comment 5: Suggested to include the DSC curved. The crystallization behavior and melting point must be presented
Response 5: The authors have addressed the reviewer’s suggestion by including the DSC thermograms in the revised manuscript (see Figure 2). Due to the overlapping of the curves in Figure 2, numerical labels for the crystallization temperature (Tc) and melting temperature (Tm) were not included directly in the plot. Instead, these values are reported in Table 3 and discussed in relation to the recycled content in lines 244–302.
Comment 6: The outcomes is not clear from the results, it is suggested to concise the conclusion.
Response 6: Thank you for the comment. The conclusion has been revised (lines 467-479) to improve clarity and conciseness. It now focuses on the key findings, such as the optimal performance of the 25%PPr blend, the predictable behavior of properties following the Law of Mixtures, the 82% drop in OIT at 100% PPr, and the 22% improvement from extrusion. These changes more clearly reflect the main results and practical implications of the study.
Reviewer 2 Report
Comments and Suggestions for Authors
The authors presented an investigation on the reuse of recycled polypropylene with virgin polypropylene, aiming at mechanical, thermal and melt flow index studies. The topic is well explored in the literature. The authors need to improve in terms of foundation with other works in the literature.
>In the abstract, authors should briefly indicate the type of processing used. In addition, add quantitative results (% percentage gains) of the best composition;
> Page2. Introduction. Authors should present a review of the specific literature on PP with recycled PP, indicating the results achieved in other reports. I recommend adding articles from 2019-2025 to address recent advances;
>In the introduction, the authors make comments and do not reinforce them with articles from the literature. For example:
“The incorporation of recycled polypropylene into virgin polypropylene…….”
“However, there are only few studies which demonstrated that incorporating low to……”
>Page 3. Authors need to organize materials and methods into topics. 2.1 Materials, 2.2 processing, 2.3 molding; and 2.4 characterization;
>Was the material dried prior to processing? Unclear.; What is the feed rate into the extruder? What is the L/D ratio in the extruder? How many heating zones are there in the extruder? Indicate the temperature in each zone. What is the profile of the screw used in the extruder? Is it configured with distribution and dispersion elements? Which ones? What is the criterion for adopting these compositions in Table 1? Is it based on the literature? Which one?
>Page 6. Authors should add melting and crystallization temperature curves; Furthermore, better support the discussion of DSC with results from the literature, especially with works on PP with recycled PP;
>Why didn't the authors add SEM morphology? This way, it would be possible to correlate morphology with mechanical properties;
>Overall, the manuscript needs to be better supported by other works in the literature on the subject, given that PP with recycled PP is widely explored.
Author Response
Dear Reviewer,
The authors sincerely thank you for your critical comments, helpful suggestions, and, most importantly, for dedicating your valuable time to review our manuscript. We have carefully considered all your comments and suggestions and believe that the revised version has been improved as a result. Please find attached a point-by-point response to your comments.
Comment1: >In the abstract, authors should briefly indicate the type of processing used. In addition, add quantitative results (% percentage gains) of the best composition;
Response 1: The abstract has been revised to include the key quantitative findings (lines 24-26) and the blends’ preparation method (line17). Specifically, the authors highlighted the change of 96% of the elastic modulus and 101% of the yield strength in the 25% PPr blend compared to PPv. These additions aim to reflect better the critical outcomes and practical implications of our study.
Comment 2: > Page2. Introduction. Authors should present a review of the specific literature on PP with recycled PP, indicating the results achieved in other reports. I recommend adding articles from 2019-2025 to address recent advances.
Response 2: The authors thank the reviewer for this helpful suggestion. In response, the Introduction section has been significantly revised to include a focused review of recent literature (2019–2025) addressing the blending of virgin and recycled polypropylene. We summarized findings from several relevant studies investigating the mechanical, thermal, and structural performance of PP/rPP blends across various blend ratios. The revised text appears on page 2 (lines 57-90).
Comment 3:>In the introduction, the authors make comments and do not reinforce them with articles from the literature. For example:
“The incorporation of recycled polypropylene into virgin polypropylene…….”
“However, there are only few studies which demonstrated that incorporating low to……”
Response 3: The authors thank the reviewer for pointing this out. These phrases were part of the earlier version of the Introduction but have since been removed during the substantial revision of that section. In the updated version, we have replaced these general statements with specific findings from recent literature (2019–2025). The revised text appears on page 2 (lines 57-90).
Comment 4: >Page 3. Authors need to organize materials and methods into topics. 2.1 Materials, 2.2 processing, 2.3 molding; and 2.4 characterization;
Response 4: We thank the reviewer for this constructive suggestion. The Materials and Methods section has been reorganized in the revised version of the manuscript to improve clarity and structure. The content is now divided into 2.1 Materials and 2.2 Methods, with additional subheadings detailing: (2.2.1), Preparation of PPv/PPr blends, (2.2.2) Preparation of tensile specimens, (2.2.3) Melt Flow Index assessment, (2.2.4) Thermal analysis, and (2.2.5) Mechanical testing. While the exact section numbering differs slightly from the reviewer's proposal, the structure follows a logical order that reflects the experimental workflow.
Comment 5: >Was the material dried prior to processing? Unclear.; What is the feed rate into the extruder? What is the L/D ratio in the extruder? How many heating zones are there in the extruder? Indicate the temperature in each zone. What is the profile of the screw used in the extruder? Is it configured with distribution and dispersion elements? Which ones? What is the criterion for adopting these compositions in Table 1? Is it based on the literature? Which one?
Response 5: The authors thank the reviewer for these detailed and helpful questions. The manuscript has been revised accordingly, and the information is now provided in lines 117–140.
Comment 6: >Page 6. Authors should add melting and crystallization temperature curves; Furthermore, better support the discussion of DSC with results from the literature, especially with works on PP with recycled PP;
Response 6: We thank the reviewer for this valuable suggestion. In response, we have expanded the thermal analysis discussion to include melting and crystallization behaviour derived from DSC measurements. Specifically, we report changes in fusion enthalpy, melting temperature, recrystallization enthalpy (Hc), and recrystallization temperature (Tc) across the PPv/PPr blends. The results are presented in Tables 3 and 4, and DSC thermograms are provided in Figure 2, showing melting and crystallization curves for all compositions.
To address the second part of the comment, the discussion has been strengthened by incorporating findings from recent and relevant literature on recycled PP blends. These additions provide a more comprehensive and literature-supported interpretation of the thermal transitions and crystallinity behaviour of virgin and recycled PP blends. The revised discussion appears on Pages 7 and 8, Lines 244–287 of the manuscript.
Comment 7:>Why didn't the authors add SEM morphology? This way, it would be possible to correlate morphology with mechanical properties;
Response 7: We thank the reviewer for this valuable suggestion. As both materials used are polypropylene, and no traces of other polymers were detected by DSC (lines 277-280), the blends were assumed to be compositionally compatible. For this reason, SEM imaging was not initially planned, as limited phase contrast is expected in miscible polymer systems. However, we acknowledge that differences in molecular weight and degradation history may still influence morphology and mechanical performance. Microstructural analysis will be considered in future work to explore these effects in more detail.
Comment8: >Overall, the manuscript needs to be better supported by other works in the literature on the subject, given that PP with recycled PP is widely explored.
Response8: We appreciate the reviewer’s general observation. In response, we have substantially expanded the literature review in the Introduction (lines 57–90) and throughout the manuscript, particularly in the thermal analysis section (lines 244–287). These revisions incorporate multiple recent studies (2019–2025) on vPP/rPP blends and now provide stronger contextual support for our findings.
Reviewer 3 Report
Comments and Suggestions for Authors
Good paper and well written.
One thing to remember is that recycled PP can have a wide spectrum of MFI’s that will make a big difference in all properties. Your vPP is a 12 MFI and your rPP is a 40 MFI. There is going to be some differences inherent in the system. Your results, mechanical and physical will change just based upon MFI and the correlated MW, and not always because it is recycled.
You discuss quite a bit about the DSC results, but there are no figures or graphs with HF curves, any reason? These should be added
Not sure what is going on in page 6, indents and font vary from the main text. Looks like they might be part of the table or equation, but it is hard to determine and a bit confusing.
Table 4 needs to have , removed and . put in place.
What OIT do you want? Just because you have a lower OIT value it does not always correlate to live of the product. Having a wide variation of MFI values would likely influence the MFI. There could also be a catalytic influence with your OIT values, ie degradation of a small amount of material could speed up the OIT.
Your TGA results contradict the OIT values, correct? You see a quicker onset of degradation with the vPP compared to the rPP. There should also be more AO’s in the virgin PP since you received commercial PP (Capilene has AOs in them already). So your agruement starting on line 315 is confusing and not really valid without knowledge of these materials.
There are quite a few liberties you have taken in your thermal analysis, TGA and OIT, that are somewhat suspect, I would reconsider other options and rephrase where appropriate.
Author Response
Dear Reviewer,
The authors sincerely thank you for your critical comments, helpful suggestions, and, most importantly, for dedicating your valuable time to review our manuscript. We have carefully considered all your comments and suggestions and believe that the revised version has been improved as a result. Please find attached a point-by-point response to your comments.
Comment 1. One thing to remember is that recycled PP can have a wide spectrum of MFI’s that will make a big difference in all properties. Your vPP is a 12 MFI and your rPP is a 40 MFI. There is going to be some differences inherent in the system. Your results, mechanical and physical will change just based upon MFI and the correlated MW, and not always because it is recycled.
Response 1: This is a very pertinent point. As clarified in the revised manuscript (lines 197–199; 215 – 220), it is explicitly stated that the differences in melt flow index (MFI) are primarily due to molecular weight differences between virgin PP and recycled PP, which includes mixed-grade sources with inherently higher MFI. This clarification addresses the reviewer’s concern that the property changes are not solely due to the recycled nature of the material.
Comment 2: You discuss quite a bit about the DSC results, but there are no figures or graphs with HF curves, any reason? These should be added
Response 2: The authors have addressed the reviewer’s suggestion by adding the DSC thermograms to the revised manuscript (see Figure 2). Because the curves in Figure 2 overlap, numerical values for the crystallization temperature (Tc) and melting temperature (Tm) were not included directly in the plot. Instead, these values are presented in Table 3 and discussed in relation to the recycled content (lines 244-302). Additionally, a hypothesis regarding the asymmetry of the recrystallization peak observed in the 25%PPr blend is provided in lines 280–287.
Comment 3: Not sure what is going on in page 6, indents and font vary from the main text. Looks like they might be part of the table or equation, but it is hard to determine and a bit confusing. Table 4 needs to have, removed and. put in place.
Response3: The authors apologize for the formatting inconsistency. The issue has been corrected in the revised manuscript. The affected section was mistakenly formatted as part of a table. It has been corrected, as shown in lines 232-240.
Comment 4: What OIT do you want? Just because you have a lower OIT value it does not always correlate to live of the product. Having a wide variation of MFI values would likely influence the MFI. There could also be a catalytic influence with your OIT values, ie degradation of a small amount of material could speed up the OIT.
Response 4: We appreciate the reviewer’s valuable observation regarding the interpretation of OIT values. We agree that a lower OIT does not necessarily provide a direct or linear correlation with the product’s lifetime, particularly when dealing with complex blends such as those containing recycled polypropylene with mixed MFI grades. In our study, OIT is used primarily as an indicator of residual antioxidant level and relative oxidative stability of the blends rather than as a precise predictor of long-term performance. The observed reduction in OIT with increasing recycled content is interpreted as evidence of partial depletion of stabilizers and possibly prior degradation during earlier processing cycles. We have clarified this point in the revised manuscript (see lines 321–325)
Comment 5: Your TGA results contradict the OIT values, correct? You see a quicker onset of degradation with the vPP compared to the rPP. There should also be more AO’s in the virgin PP since you received commercial PP (Capilene has AOs in them already). So your agruement starting on line 315 is confusing and not really valid without knowledge of these materials.
Response 5: Thank you for this critical observation. The authors agree that the interpretation of the thermal degradation behaviour requires clarification, particularly the apparent discrepancy between the TGA and OIT results. As noted, commercial virgin PP (Capilene) likely contains standard antioxidants. However, as revised in lines 381–383 we clarify that recycled PP may retain residual stabilizers from its original applications, which can delay thermal degradation under inert conditions. A clear distinction is now made between thermal stability (assessed by TGA under nitrogen) and oxidative stability (measured by OIT), and the apparent contradiction is explained. The updated text reflects these clarifications in lines 347 – 402.
Comment 6: There are quite a few liberties you have taken in your thermal analysis, TGA and OIT, that are somewhat suspect, I would reconsider other options and rephrase where appropriate.
Response 6: The authors thank the reviewer for this valuable feedback. In response, we have carefully reviewed the results throughout the manuscript. Where appropriate, we propose plausible explanations and acknowledge the limitations of the current methods, particularly in the absence of detailed compositional knowledge of the recycled polymers.
Round 2
Reviewer 1 Report
Comments and Suggestions for Authors
All the comments were addressed, and the manuscript can be accepted in its current form.
Reviewer 2 Report
Comments and Suggestions for Authors
The authors responded satisfactorily to the questions, further improving the quality of the manuscript. Recommendations were reviewed and added to the new version of the article, which makes publication possible.